**METHODS AND PROTOCOLS**
Molecular Biology and Physiology

# Benchmarking Bacterial Promoter Prediction Tools: Potentialities and Limitations

Murilo Henrique Anzolini Cassiano,ᵃ  Rafael Silva-Rochaᵃ

ᵃFMRP - University of São Paulo, Ribeirão Preto, SP, Brazil

**ABSTRACT**  The promoter region is a key element required for the production of RNA in bacteria. While new high-throughput technology allows massively parallel mapping of promoter elements, we still mainly rely on bioinformatics tools to predict such elements in bacterial genomes. Additionally, despite many different prediction tools having become popular to identify bacterial promoters, no systematic comparison of such tools has been performed. Here, we performed a systematic comparison between several widely used promoter prediction tools (BPROM, bTSSfinder, BacPP, CNNProm, IBBP, Virtual Footprint, iPro70-FMWin, 70ProPred, iPromoter-2L, and MULTiPly) using well-defined sequence data sets and standardized metrics to determine how well those tools performed related to each other. For this, we used data sets of experimentally validated promoters from *Escherichia coli* and a control data set composed of randomly generated sequences with similar nucleotide distributions. We compared the performance of the tools using metrics such as specificity, sensitivity, accuracy, and Matthews correlation coefficient (MCC). We show that the widely used BPROM presented the worse performance among the compared tools, while four tools (CNNProm, iPro70-FMWin, 70ProPred, and iPromoter-2L) offered high predictive power. Of these tools, iPro70-FMWin exhibited the best results for most of the metrics used. We present here some potentials and limitations of available tools, and we hope that future work can build upon our effort to systematically characterize this useful class of bioinformatics tools.

**IMPORTANCE**  The correct mapping of promoter elements is a crucial step in microbial genomics. Also, when combining new DNA elements into synthetic sequences, predicting the potential generation of new promoter sequences is critical. Over the last years, many bioinformatics tools have been created to allow users to predict promoter elements in a sequence or genome of interest. Here, we assess the predictive power of some of the main prediction tools available using well-defined promoter data sets. Using *Escherichia coli* as a model organism, we demonstrated that while some tools are biased toward AT-rich sequences, others are very efficient in identifying real promoters with low false-negative rates. We hope the potentials and limitations presented here will help the microbiology community to choose promoter prediction tools among many available alternatives.

**KEYWORDS**  promoter prediction, bacterial promoters, *cis*-regulatory elements, bioinformatics, promoter prediction

Address correspondence to Rafael Silva-Rocha, silvarochar@usp.br.

Do you want to predict bacterial promoter and do not know which tool to use? You really need to read this!

Promoter regions are intrinsic DNA elements located upstream of genes and required for their transcription by the RNA polymerase (RNAP) (1). Thus, the correct mapping of promoters is a critical step when studying gene expression dynamics in bacteria. While the definition of promoters could vary widely, here we will consider promoters as the core elements recognized by the sigma subunit of the RNAP. In *Escherichia coli*, seven alternative sigma factors are responsible for gene expression,

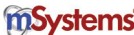

while sigma 70 is the most important one as it is required for the expression of housekeeping genes (2, 3). Therefore, this sigma factor recognizes a consensus region about 35 bp in length with two key elements, the −10 box (with consensus motif TATAAT) and the −35 box (TTGACA) which are separated by $17 \pm 2$ bp (1, 2). In addition to the core promoter region, other *cis*-regulatory elements can play relevant roles in the regulation of gene expression (4). In this sense, the production of RNA at the transcription start site (TSS) is the result of the interplay between the core promoter region and the *cis*-regulatory elements (5). Mapping of functional promoter elements have been performed mostly using low-throughput techniques (such as promoter probing, primer extension, DNA footprinting, etc.) or more recently, by high-throughput experimental approaches (RNA-seq [high-throughput RNA sequencing], SELEX [genomic systematic evolution of ligands by exponential enrichment], Sort-seq [flow cytometry, sorting, and next-generation sequencing], etc.) (6–11). However, the rapidly growing number of fully sequenced bacterial genomes greatly exceeds our ability to map promoter elements experimentally. Therefore, diverse computational tools have been created to predict promoters/TSSs at specific genes or genomic levels.

Some of the first approaches to map promoters have been based on the use of position weight matrices (PWMs) of −10 and −35 box motifs, taking into account the distribution of the spacer length between the motifs and their distance from TSSs (12, 13). Yet, over the past years, a growing number of computational strategies have evolved in complexity. Notable novel approaches raised, such as sequence alignment-base kernel for support vector machine (14, 15), profiles of hidden Markov models combined with artificial neural networks (16), or weighted rules extracted from neural network models (17). Also, new ways to extract information from DNA sequences to perform predictions have appeared. Thus, there are now several numerical representations of DNA sequences in which each one carries its properties (18–20), such as methods that use k-mer frequencies or variations (21, 22) and other methods that include physicochemical properties of DNA (23).

Recently, machine learning (ML) techniques have been used to obtain insight from different sources from diverse biology fields (an extensive survey can be seen in Libbrecht and Noble [24], Camacho et al. [25], and Zou et al.[26]), and in the past few years, this has been applied to the recognition of promoters, TSSs, and regulatory sequences. Among most of the ML algorithms used for this purpose, we can mention support vector machine (27), neural networks (28), logistic regression (29), decision trees (30), and hidden Markov models (31, 32). Despite the existence of all these modern techniques, promoters cannot always be inferred based on their sequence only, and currently, we have no clue on how efficient these tools are. This occurs since each new tool is validated without the use of standardized data sets or methods, making it difficult to compare novel emerging alternatives with the current state of the art. In this work, we summarize general aspects of the available promoter prediction tools, exposing comparatively their main strong and weak features. For this, we compared the performance of these tools using experimentally validated promoters from *E. coli*. Unexpectedly, we show that some very popular tools such as BPROM performed very poorly compared to tools created over the last 2 years. We hope our results can help both community users to choose a suitable tool for their specific applications, as well as developers to construct novel tools overcoming key limitations reported here.

## RESULTS AND DISCUSSION

**Describing the tools: methods, availability, and usability.** In this section, we present a succinct explanation of each methodology (see Table 1) as well as the usability information about their use requirements, acceptable file types, etc. (see Table 2). Below, we describe briefly for each tool how they have been built and some of the main features.

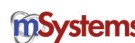

**TABLE 1** General information on the tools used here

| Tool | Method | Training sequence data set[a] | No. of E. coli sigma factors | Availability | Yr | Reference | No. of citations (Google Scholar)[b] |
|---|---|---|---|---|---|---|---|
| BPROM | Weight matrices of different motifs combined with linear discriminant analysis | Positive: Experimentally validated promoters from E. coli (14). Negative: Inner regionsof protein-coding ORFs. | 70 | Web server http://www.softberry.com/berry.phtml?topic=bprom&group=programs&subgroup=gfindb | 2011 | 33 | 427 |
| bTSSfinder | Position weight matricesfor promoter elements, oligomer frequencies, physicochemical properties as features, and Mahalanobis distance for feature selection and with neural network for classification | Positive: Experimentally validated TSSs from Regulon DB. [−200, +51]. Negative: Genomic regions withno experimental evidence for the presence of TSSs. | 24, 28, 32, 38, 70 | Stand-alone and Web server http://www.cbrc.kaust.edu.sa/btssfinder/ | 2016 | 23 | 26 |
| BacPP | Weighted rules extracted from neural network | Positive: Regulon DB available promoters. [−60, +20]. Negative: randomly generated sequences (with established nucleotide frequencies) and intergenic sequences. | 24, 28, 32 38, 54, 70 | Web server http://www.bacpp.bioinfoucs.com/home | 2011 | 17 | 22 |
| Virtual Footprint | PWMs from different available databases | | | Web server http://www.prodoric.de/vfp/vfp_promoter.php | 2005 | 36 | 370 |
| IBBP | Image-based and evolutionary approach which generates "images" (template-image strings that keep features of spatial sequence relationships) | Positive: sigma 70 promoters from Regulon DB. [−60, +20]. Negative: randomly generated from protein-coding sequences. | 70 (expandable approach) | Source code https://github.com/hahatcdg/IBPP | 2018 | 35 | 1 |
| iPro70-FMWin | 22,595 features extracted from sequence and AdaBoost to select the most representatives among then; logistic regression classifier | Positive: Regulon DB annotated promoters. [−60, 20]. Negative: randomly generated from protein- coding and intergenic region sequences. | 70 | Webserver http://ipro70.pythonanywhere.com/ | 2019 | 38 | 4 |
| 70ProPred | Support vector machine using position-specific tendencies of trinucleotide and electron-ion interaction pseudopotentials as features | Positive: promoters from Regulon DB. [−60, 20]. Negative: randomly generated from coding and noncoding sequences. | 70 | Webserver http://server.malab.cn/70ProPred/ | 2017 | 39 | 33 |
| CNNProm | Convolutional neural networks | Positive: promoters from Regulon DB. [−60, 20]. Negative: the opposite chain of randomly selected protein-coding genes. | 70 | http://www.softberry.com/berry.phtml?topic=index&group=programs&subgroup=deeplearn | 2017 | 34 | 60 |
| MULTiPly | Support vector machine using biprofile Bayes, KNN features, k-tuple nucleotide compositions, and dinucleotide-based auto-covariance as features | Positive: promoters from Regulon DB. [−60, 20]. | 24, 28, 32, 38, 54, 70 | Web server and stand-alone http://flagshipnt.erc.monash.edu/MULTiPly/ | 2019 | 41 | 30 |
| iPromoter-2L | Multiwindow-based pseudo k-tuple nucleotide composition with physicochemical properties as features and Random Forest as a predictor | Positive: promoters from Regulon DB. [−60, 20]. Negative: randomly extracted from the middle regions of long coding sequences and convergent intergenic region (none of the promoters in each set has more than 0.8 pairwise sequence identity) | 24, 28, 32. 38, 54, 70 | Web server http://bioinformatics.hitsz.edu.cn/iPromoter-2L/ | 2018 | 40 | 180 |

[a]Positive, positive sequences, sequences expected to be promoters. Negative, negative sequences, sequences expected to not include promoters. The interval of the sequence with the boundary numbers related to a TSS is indicated within brackets ([−60, +20], [−60, +19], [−60, +20], or [−200, +51]).
[b]Citations checked on 3 May 2020.

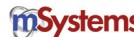

**TABLE 2** Usage characteristics of the tools analyzed here

| Tool | Multifasta | Big files | Shows promoter core | Score or probability | Uppercase only | Output format | Execution time | Follow up | Interface | Comment(s) |
|---|---|---|---|---|---|---|---|---|---|---|
| BPROM | No | Yes | Yes | Yes | No | Text on screen | Fast | Progress on screen | Webform, simple, intuitive | Multifasta not supported and sequence boxes are not shown; difficult to process the results |
| bTSSfinder | Yes | Yes | Yes | Yes | No | Text file, GFF file, BED file | Fast | Progress on screen | Login needed, webform, simple, intuitive | Flexible configurations of cutoff values; results saved for 1 week; Linux tool available for download; it needs a large promoter sequence (−200, +50, related to the putative TSS) |
| BacPP | Yes | No | No | Yes | No | Text on screen or text file | Fast | N | Login needed, webform, simple, intuitive | Short tests per time |
| Virtual Footprint | No | Yes | Yes | Yes | No | Text on screen | Medium fast | Progress on screen | Webform, many fields, and option in the screen | Integrated with a large PWM database of TFBS; applicable to many species; limited to the position weight matrix available |
| IBBP | No | Yes | It shows the putative TSS | Yes | No | Text file | Fast | Progress on screen | Command line | Windows SO only execution; requires the manual input files; training and test procedures are separated; fast for big files; it can be used as an approach to the initial prediction of any type of promoters |
| iPro70-FMWin | Yes | Yes | No | Yes | No | Text on screen | Fast | No | Webform, simple, intuitive | High accuracy |
| 70ProPred | Yes | Yes | No | No | Yes | Text on screen | Fast | No | Webform, simple, intuitive | High accuracy; it does not accept a file as input, just text on a form |
| CNNProm | Yes | Yes | No | Yes | Yes | Text on screen | Long time (for genomes), fast for multifasta | No | Webform, simple, intuitive | Useful for large sequences (genomes); without a follow-up display; multifasta not supported and sequence boxes are not shown; difficult to process the results |
| MULTiPly | Yes | Yes | No | No | No | Text on screen or text file | Medium time | Progress on screen, job ID to find the result later | Webform, simple, intuitive | Good accuracy; it saves the result; time-consuming for large sequences |
| iPromoter-2L | Yes | Yes | No | No | No | Text on screen | Fast | Progress on screen | Webform, simple, intuitive | Good accuracy |

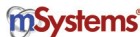

**(i) BPROM.** BPROM (33) was developed as a module of an annotation pipeline for microbial sequences to find promoters in upstream regions of predicted open reading frames (ORFs). To train the model, the authors used a data set of experimentally validated promoters from elsewhere (14). They applied linear discriminant analysis to discriminate between those promoters and inner regions of protein-coding sequences. For attributes, they used five position weight matrices of promoter conserved motifs and they also consider the distance between the −10 and −35 boxes and the ratio of densities of octanucleotides overrepresented in known bacterial transcription factor binding site (TFBS) relative to their occurrence in coding regions. This tool is available as a web application, and users can submit a local file or paste the sequence in the web form. It quickly returns the results in the screen with the possible −10 and −35 boxes of predicted promoters and their positions in the submitted sequence.

**(ii) bTSSfinder.** bTSSfinder (23) is a tool that predicts putative promoters for different sigma factors in *E. coli* and cyanobacteria. Its positive data set consists of experimentally validated *E. coli* TSSs from Regulon DB and different experimentally mapped cyanobacterial TSSs provided by several works. Its negative data set consists of genomic regions where there is no experimental evidence for the presence of TSSs. They started with 30 features distributed between these types: promoter element motifs (PWMs), the distance between the elements, oligomer scores, TFBS density, and physicochemical properties. The final set of features was selected by evaluating the predictive power of these features by calculating Mahalanobis distance and used to train a neural network. This tool is available as a web application or as a stand-alone tool for Linux. On the website, an e-mail is needed to login and the results are saved for a week.

**(iii) BacPP.** BacPP (17) is a prediction tool to find *E. coli* and other *Enterobacteriaceae* promoters. For a positive data set, the authors used promoter sequences from Regulon DB for six different sigma factors in *E. coli* and other *Enterobacteriaceae* promoter sequences obtained from several works. For its negative data set, they used two approaches: (i) random sequences generated with a probability of 28% for nucleotides adenine and thymine and 22% for cytosine and guanine; (ii) random selected intergenic regions. Each nucleotide of these sequences was transformed into binary digits and used to train neural networks. To use this tool, the user must create a login in the website, then paste the sequences or fasta file according to their model, and select the sigma factors of interest.

**(iv) CNNProm.** CNNProm (34) is a web tool that can predict prokaryotic and eukaryotic promoters from big genomic sequences or multifasta files. In the case of *E. coli* promoters, the authors took the sequences from Regulon DB, and the negative controls (nonpromoter sequences) were randomly selected from the opposite chain of coding regions in genomes. Each of these sequences was transformed into a binary four-dimensional vector and used directly as features to train a convolutional neural network. To use this predictor, users must enter the sequences or the file on the website and choose the organism model.

**(v) IBBP.** IBBP (35) is a stand-alone application that implements a new approach called "image-based promoter prediction." This approach consists of generating multiple "images": template strings carrying possible features/elements presented in promoters and their spatial relationships. The image generation and selection are conducted by applying an evolutionary approach and calculating the similarity of these images in a set of *E. coli* sigma 70 promoters. The authors measured the accuracy of the tool by analyzing the set of promoters and protein-coding sequences. To use this software, it is necessary to download the executable files, execute the evolutionary algorithm with the promoters of interest, and then implement the classifier software, which uses the resulting model generated in the previous step.

**(vi) Virtual Footprint.** Virtual Footprint (36) is a web framework for prokaryotic regulon prediction. This framework makes use of several PWMs provided by PRODORIC (37) and other PWMs from other sources. To make the prediction, it is necessary to

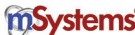

upload a DNA sequence or a fasta file, select different PWMs for core promoter elements or other transcription factor binding sites, and set some parameters.

**(vii) iPro70-FMWin.** iPro70-FMWin (38) is a web application for sigma 70 promoter prediction. Its training data set consists of sigma 70 promoter sequences from data set Regulon DB 9.0 and sequences randomly chosen from coding and intergenic regions of *E. coli* as positive and negative data sets, respectively. For feature extraction, 22,595 sequence-based features were generated for "multiple windows," i.e., different regions of the promoter sequence. These features include, for example, different kinds of k-mer and g-gapped k-mer compositions and statistical and nucleotide frequency measures. Among the machine learning methods tested by the authors, logistic regression achieved better results. They also applied the AdaBoost technique for feature selection to improve prediction.

**(viii) 70ProPred.** 70ProPred (39) was built using sigma 70 promoter sequences from Regulon DB 9.0 and randomly generated sequences from coding and noncoding regions of the *E. coli* genome to train a support vector machine (SVM) model. The attributes generated from the sequences were position-specific trinucleotide propensity and electron-ion interaction pseudopotentials of nucleotides, considering single- or double-stranded DNA, to reveal trinucleotide distribution differences between the samples and represent the interaction of trinucleotides, respectively.

**(ix) iPromoter-2L.** iPromoter-2L (40) is an online tool that provides the prediction for all *E. coli* sigma promoters. This method has two "layers" of classification applying random forests; first, it resolves whether a given sequence is a promoter, and then it selects the sigma factor class. For model training, the authors used experimentally confirmed promoter sequences from Regulon DB 9.3 as the positive data set, and randomly extracted sequences from the middle regions of long coding sequences and convergent intergenic regions as the negative data set. It is important to emphasize that sequences with more than 0.8 pairwise sequence identity for a given sigma factor promoter data set were removed to reduce identity biases. Their feature extraction was based on multiwindow-based pseudo K-tuple nucleotide composition, which consists of a sliding window, extracting and encoding physicochemical attributes of different regions of a given sequence.

**(x) MULTiPly.** The MULTiPly (41) web application provides promoter prediction for all *E. coli* sigma factors. To train their model, the authors used experimentally validated promoter sequences from Regulon DB for all type of sigma factors in *E. coli*. Their feature extraction was divided into two types; the first one was used to represent global features, applying biprofile Bayes and KNN (k-nearest neighbor) features, and the second one was used to represent local features, applying k-tuple nucleotide composition (sequence-based feature) and dinucleotide-based auto-covariance (which considers physicochemical properties). This method also performs two steps of classification: first, it resolves whether a given sequence is a promoter or not, and then it decides to which class of sigma promoter it belongs. The authors used the SVM method for classification and the F-score method for feature selection.

These last four web tools (iPro70-FMWin, 70ProPred, iPromoter-2L, and MULTiPly) are used in similar ways, accepting multifasta formatted sequences on a simple web form and returning the results on the screen. Information and characteristics of the tools and a summarization of the approaches discussed above are presented in Tables 1 and 2.

**Analyzing the performance of promoter prediction tools.** In order to compare the performance of the promoter prediction tools presented above, we analyzed the positive and negative data sets as described in Materials and Methods. From the 10 algorithms selected, BacPP could not be tested with our entire data set, because multifasta files were not supported, and Virtual Footprint produces a large number of predicted −10 boxes for sigma 70 in both positive and negative data sets, a number that greatly exceeds the number of sequences analyzed. Thus, these two tools were not considered in further analyses. Of the remaining eight algorithms, five achieved more

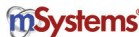

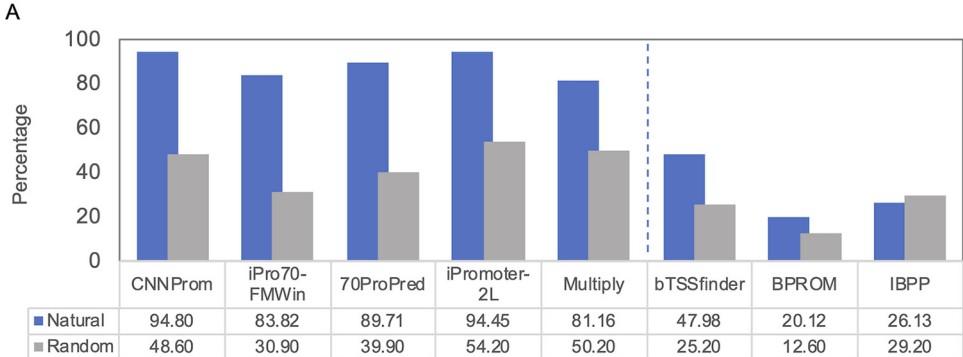

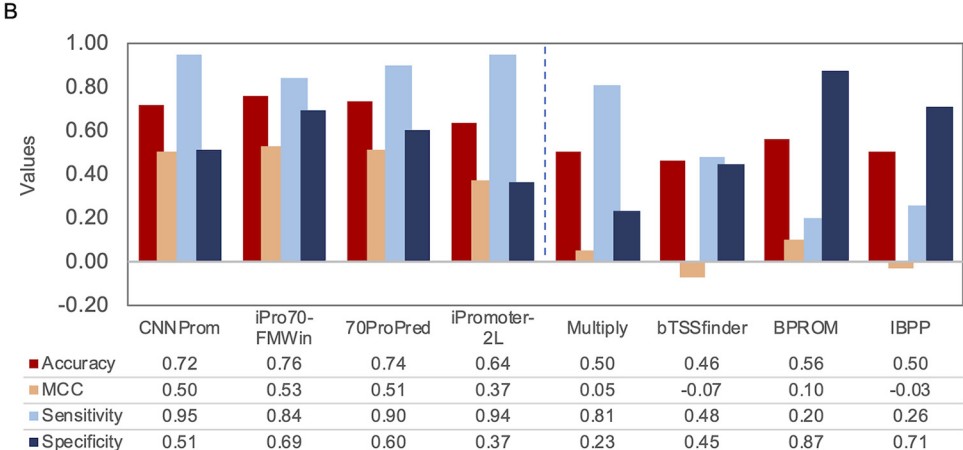

**FIG 1** Analysis of the performance of promoter prediction tools. (A) Percentage of sequences predicted as sigma 70-dependent promoters in both data sets. The percentage of correct classifications of experimental promoters (blue) and the percentage of misclassified random sequences (gray) are presented. The vertical dashed line separates the five best tools from the three worse tools analyzed. (B) Metrics used to evaluate the performance of the tools. Note that MCC values range from −1 to 1. It is important to emphasize that two tools presented the highest sensitivity associated with low specificity, i.e., tools usually perform good classifications for real promoters and high misclassification of random sequences. The vertical dashed line divides the four best tools from the four worse tools.

than 50% of correct classification on the positive data set, while six correctly classified 50% of the negative data set (Fig. 1A). The best performance was observed for CNNProm (94.8% true positive [TP]), followed by iPro70-FMWin (94.5% TP), 70ProPred (89.7% TP), iPromoter-2L (83.8% TP), and MULTiPly (81.2% TP). When we compared the performance parameters (accuracy, Matthews correlation coefficient [MCC], sensitivity, and specificity), we observed that four of them (CNNProm, iPromoter-2L, 70ProPred, and iPro70-FMWin) presented the best performance, while MULTiPly scores high only for sensitivity (Fig. 1B). Therefore, we can observe MCC values close to zero for the remaining four tools (MULTiPly, bTSSFinder, BPROM, and IBPP), indicating that these tools performed close to random classifications. It is interesting to notice that BPROM, a widely cited and used tool, presented the worst results together with bTSSFinder and the IBBP, but also presented fewer false-positive (FP) results. We also found that IBBP's method based on the evolutionary approach classified random sequences as promoters more often than in the real promoter data set (i.e., it displays a higher FP rate than TP rate). From the analysis presented in Fig. 1, we can observe that iPro70-FMWin performed best due to a small number of FP results and the overall best results of all metrics used (Fig. 1B).

Next, we performed a hierarchical clustering analysis using the results from the five tools that presented the best results. As can be seen in Fig. 2A for the positive data set, results obtained with iPromoter-2L were more correlated with CNNProm outcomes,

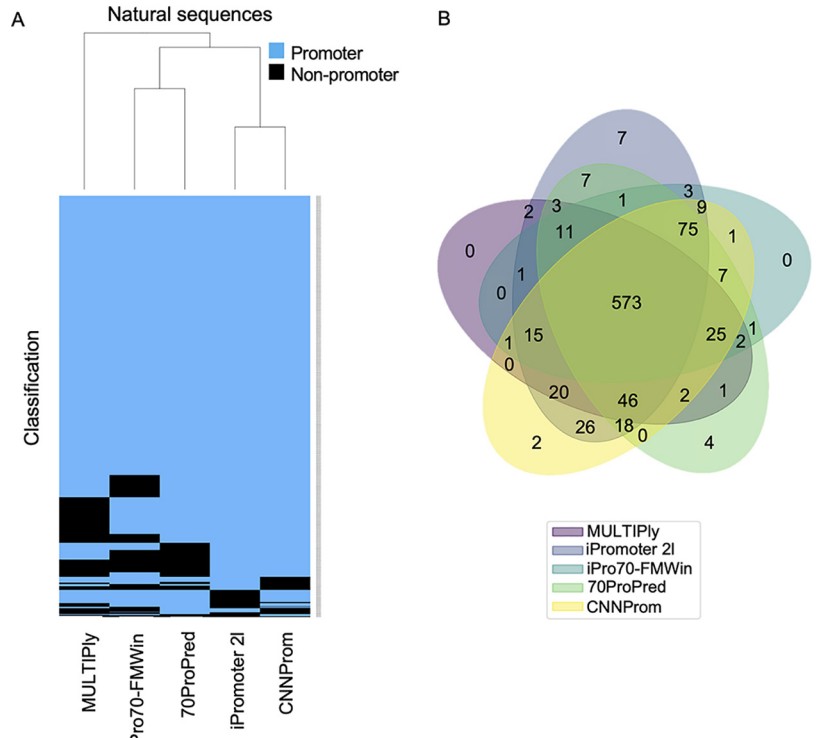

**FIG 2** Analysis of tool performance in the positive data set (natural sequences). (A) Hierarchical clustering of DNA sequences classified as promoters (blue) or nonpromoters (black). (B) Venn diagram representing the number of sequences predicted as promoters from panel A.

since both produced the largest sets of TP predictions, while iPro70-FMWin was more related to 70ProPred. In general, 573 sequences (62.2%) were correctly classified by all five algorithms (Fig. 2B). When we analyzed the negative data set (constructed with random sequences), we do not observe a clear clustering since each tool presented a different level of FP results, with the lowest level observed for iPro70-FMWin (Fig. 3A). In this case, only 102 sequences (10.2%) were incorrectly classified as a promoter by all tools, indicating that each algorithm has specific features to equivocally classify the random sequences. It is worth mentioning that the three best tools (CNNProm, iPro70-FMWin, and 70ProPred) are from 2017 to 2019, indicating that, as expected, promoter prediction algorithms are evolving through the years. Taken together, these results indicate that four out of eight tools analyzed here display equivalent predicting power to identify true promoter sequences, while the widely used tool BPROM exhibits a reduced predictive capability.

**Identification of promoter features identified during the analyses.** As presented above, we observed a high degree of similarity between the best tools for the identification of true promoters, but a lower overlap on random sequences equivocally classified as promoters. This could indicate that each algorithm might identify different features to assign a sequence as a promoter. To further investigate this process, we analyzed the information content from the sequences identified as promoters from the positive and negative data sets for the top five tools analyzed here. The results of these analyses are presented as sequence logos in Fig. 4 and 5 for the positive and negative data sets, respectively. As can be seen in Fig. 4, TP sequences identified by all five algorithms display the same consensus sequence that resembles a strong canonical −10 box from sigma 70 promoters (2). It is worth noticing that the information content was higher for iPro70-FMWin (up to 0.4 bits), which also displayed the best performance according to the metrics used here. However, when we analyzed the data from promoters identified in the random sequences, we could see a much fuzzier signal for

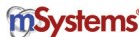

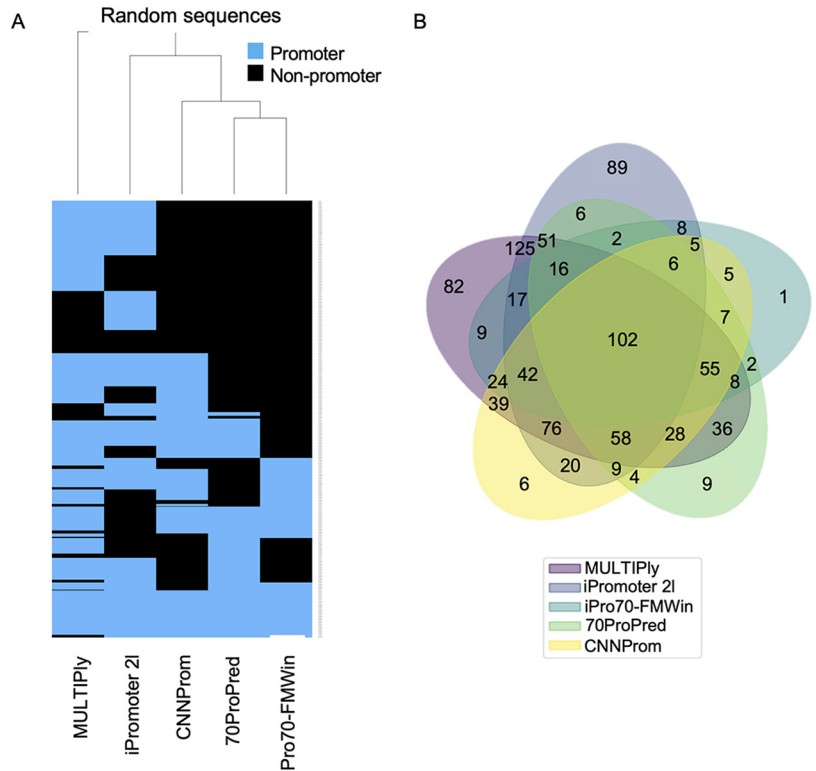

**FIG 3** Analysis of tool performance on the negative data set (random sequences). Hierarchical clustering of DNA sequences classified as promoters (blue) or nonpromoters (black). (B) Venn diagram representing the number of sequences predicted as promoters from panel A.

MULTiPly, iPromoter-2L, and CNNProm, which were the three tools with the highest FP rate from the top five tools (Fig. 5), indicating that the rich A (adenine) and T (thymine) frequencies play a role on false-positive classification. We also could observe that, in the case of random sequences predicted as promoters for all algorithms, we obtained a more evident −10 box motif and it still shows high A and T influences (see Fig. S1 in the supplemental material). This implies that these tools are sensitive to AT content, which makes sense since iPromoter-2L and CNNProm were trained on coding sequences as negative controls (34, 40). On the other hand, 70ProPred and iPro-70FMWin, which presented the lower FP rate, presented clearer −10-like signals similar to those identified on the positive sequences, although with lower information content. This might be explained by these tools classifying sequences that resemble true promoters, and we could not rule out the possibility that some of these random sequences could in fact display promoter activity in *E. coli* if tested experimentally. Taken together, these results indicate that high rates of FP results observed for some of these algorithms could be due to the use of unrealistic control sequences (such as coding regions) that could make the algorithms sensitive to AT-rich regions, highlighting the importance of choosing appropriate nonpromoter sequences to train these tools.

**Conclusions.** In this work, we performed a benchmark analysis of the performance of promoter prediction tools using a well-characterized promoter sequence and random sequences. As can be seen from the results above, new tools have emerged with enhanced performance compared to widely used ones. Although the best performing tool uses just sequence-based features (a result that corroborates with Abbas et al. [42]), in general, algorithms using feature extraction that combines attributes derived from sequence together with physicochemical properties of DNA achieved better results. It is also clear from our results that choosing the appropriate control (or negative) data set to construct these algorithms is crucial to avoid false-positive results. Therefore, coding sequences or sequences with different features render the tools AT

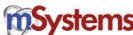

**FIG 4** Analysis of the information content of DNA sequences identified as promoters on the positive data set (natural sequences). The sequence logos are shown for sequences predicted as sigma 70-dependent promoters by MulTiPly (A), iPromoter-2L, (B) iPro70-FMWin (C), 70ProPred (D), and CNNProm (E).

sensitive, increasing the false-positive rate. Furthermore, we still need an experimentally well-validated nonpromoter data set to faithfully use as negative controls in these predictions, but these sequences are not available yet. In this sense, we expect that the growing number of high-throughput experiments could become a great source of data to create novel data sets to train new tools for promoter prediction in the future. Another complication to this subject comes from recent evidence showing that just one mutation in random sequences could lead to constitutive transcription *in vivo*, indicating that transcription is indeed a robust process (43). Therefore, future attempts have

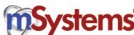

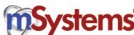

**FIG 5** Analysis of the information content of sequences identified as promoters on the negative data set (random sequences). The sequence logos are shown for DNA sequences predicted as sigma 70-dependent promoters by MulTiPly (A), iPromoter-2L (B), iPro70-FMWin (C), 70ProPred (D), and CNNProm (E).

to be made to create complete data sets with very similar promoter/nonpromoter sequences to train next-generation tools.

Additionally, several sources of prior information could be incorporated into prediction methods to improve the final tools. For instance, the interrelation between the UP (upstream promoter) element and a subunit of RNAP was found to play a role in transcription initiation and promoter activity (44) and switch preference of sigma factors in promoters (45). Besides, specific nucleotide composition and motifs between −10 and −35 boxes leading to different DNA curvatures were found to influence transcription initiation and promoter activity (46, 47). Additionally, more than 300 proteins in *E. coli* are predicted to bind DNA, and half of them have their function

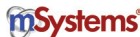

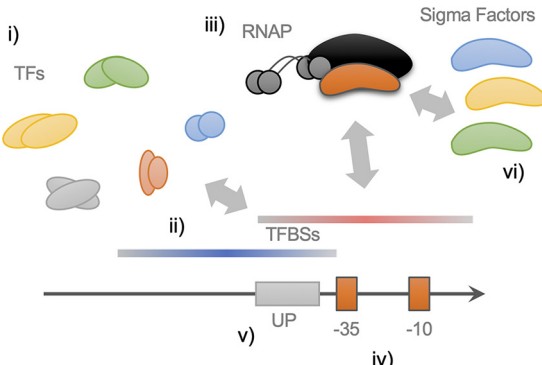

**FIG 6** A putative model for a bacterial promoter region, including a range of experimental attributes. (i) More than 300 proteins (transcription factors [TFs]) in *E. coli* are predicted to bind DNA, and there is a lack of experimental characterization (1). (ii) Recently, high-throughput studies, such as genomic SELEX, are showing a large number of possible TFBS on genomes (11), which may impact on the composition of promoter sequences. These regions can have positive (blue regions) or negative (red regions) effect on promoter activity. (iii) RNAP requires a sigma factor to be recruited to the promoter sequence, and each sigma factor possesses a preference for a specific motif on DNA (1). (iv) Nucleotide composition and motifs between −10 and −35 boxes influence transcription initiation and promoter activity (46, 47). (v) The interrelation between the UP element and a subunit of RNAP were found to play a role in transcription initiation and promoter activity (44), and the UP element can switch preference of sigma factors in promoters (45). (vi) The same promoter sequence can respond to diverse sigma factors, according to experimental characterizations and *in silico* approaches (36, 61, 62).

experimentally characterized (1). These proteins could thus impact promoter activity *in vivo*, and their binding sequence preferences could influence promoter discovery. Finally, recent studies with genomic SELEX show that the number of transcription factor binding sites (TFBS) annotated in databases is underestimated (11). , it is worth noticing that a promoter is a complex entity that requires a large number of elements, and therefore, the transcription observed *in vivo* for a specific DNA element could be due to several interacting factors which perhaps could not be predicted using a single tool (Fig. 6).

A notable characteristic shared by the works mentioned here is that all the available prediction tools perform only binary classifications, i.e., not considering whether a data set of promoters contains constitutive or regulated promoters. Therefore, there is no indication of an activity threshold to classify a given sequence as a promoter, and it is known that expression levels of different bacterial transcripts vary on a wide range of magnitude order (48). However, there have been some attempts in the literature to perform some regression analysis instead of binary classification only. For instance, De Mey and colleagues (49) synthesized a library of 57-bp-long sequences designed to have conserved, semiconserved, and random nucleotides and the perfect consensus of −35 and −10 boxes. They also added variance in sequences that surround the core promoter and that may play a role in promoter activity. Performing a fluorescence assay to measure promoter activity and applying a partial least squares regression model, they attempt to predict promoter strength for sigma 70. Interestingly, their results found no correlation between promoter strength and anomalies in the spacer sequence length or the −10/−35 boxes (49). Yet, only 78 variants were characterized in this experimental design, and more variants are needed to train an accurate model. Similarly, Rhodius et al. (50) used 60 promoters for the alternative sigma E, extracting as attributes, PWMs for different motifs: −35, spacer, −10, discriminator, start, and initial transcription region. For the far-distal, distal, and proximal motifs, which are components of the UP element, they constructed a model to extract A/T content and A- and T-tract length/frequency. Also, a spacer and discriminator length penalty score was added. Notably, *in vivo* and *in vitro* expression was measured in their work, and promoter activity was also tested by a function of sigma E concentration. Additionally, partial least squares regression was used to predict promoter activity (50). This approach is useful to find the elements in a given promoter sequence, and by using

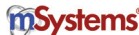

cross-validation, the results appear to be promising, despite the small size of the data set and the use of PWM (a model that showed poor predictive results on our work). Moreover, instead of using position weight matrices, energy matrices are being successfully built to represent the sequence-dependent binding energy using sequence libraries with a large number of variants followed by Sort-seq experiments (flow cytometry, sorting, and next-generation sequencing) (6), and therefore, these energy matrices are being employed to model promoter activity (51). Urtecho et al. (52) assembled a large library containing 12,288 variants of sigma 70 promoters, composed of one of eight defined sequences of −35 boxes, −10 boxes, spacers, backgrounds, and three defined UP elements to understand the contributions and relations of the *cis* elements on promoter sequences. The authors have integrated their expression cassette on different genomic locations and have investigated its effects, applying a well-suited method for expression normalization. Therefore, their approach explained most of the variance in promoter activity, as well as discovered nonlinear interactions between promoter elements by employing neural networks (52). Despite the limitation of a data set with discrete characteristics, this approach presents a reliable method to predict promoter strength in a well-defined context, but application of these methods to natural systems has still to be demonstrated. In the future, approaches to find a promoter followed by regression models to predict its activity/strength need to be publicly available to allow the community to reach a better framework for metabolic engineering and other applications using synthetic biology (53).

One final remark is that the majority of algorithms have been created using data sets of promoters from just one bacterium, *E. coli*. Consequently, since each organism has its particularities in terms of DNA binding proteins and sigma factor elements, we are still far away from having a prediction tool that can be used for several organisms. To accomplish that, we would require extensive promoter data sets from several microorganisms to construct multipurpose prediction tools. Last, we hope the approach and metrics used here can contribute to future studies aimed to construct improved promoter prediction tools.

## MATERIALS AND METHODS

**Selecting promoter prediction tools.** We started this work by searching in the literature for recent and available prediction tools for *E. coli* promoters. For each case, when a tool was available online or by software download, we selected it for posterior analysis. Table 1 shows the summarized information about the tool methodology (i.e., implementation, the approach used, or process performed), the sigma factors it can predict, the available format, and the access links. All these descriptions have been extracted from the original papers describing the tools. Next, we analyzed some usability features of the tools (such as the file format accepted as input, maximal allowed file size, the output format of results, etc.) as summarized in Table 2. Then, we selected the ones that accepted our complete data set in multifasta format as input to perform a comparative analysis.

**Promoter data sets used for the analyses.** To compare each selected tool, we used an experimentally validated promoter data set for the well-studied *E. coli* K-12 which is dependent on sigma 70, as available in the curated database Regulon DB 10.5 (54). We used only sigma 70-dependent promoters since they are mostly well-characterized in bacteria, and consequently, most tools have been developed to recognize this class of elements. Thus, our so-called positive data set was formed by 865 natural sequences extracted from Regulon DB and classified as having a strong evidence/confidence level. Additionally, we used a negative promoter set consisting of 1,000 randomly generated sequences with a nucleotide distribution similar to that encountered in the 865 natural sequences, which was constructed with an *ad hoc* script written in Python. We chose this strategy for two reasons: (i) generating a negative data set with this approach allows us to assess the tool's capacity to distinguish real promoters from random sequences, and (ii) to the best of our knowledge, there is no experimentally validated negative promoter data set available. Also, it is important to stress that many tools, such as BPROM, 70ProPred, and iPro-70FMWin, used coding and intergenic regions as control (negative) sequences, but this is not appropriate since coding and noncoding regions have different nucleotide compositions and structural properties (55, 56). In our data sets, the sequences have 81 bp, since most tools consider and require as input 60 bp upstream and 20 bases downstream of the putative TSS (the region interval [−60, +20]). In the case where the tool required the entire genome, we used the *E. coli* K-12 MG1655 genome (GenBank accession no. U00096.3) and when a tool required a bigger interval than [−60, +20] bp, we extracted the additional sequence from this same genome. The two data sets (natural and random) used here are available as Data Sets S1 and S2 in the supplemental material.

**Building the negative data set.** The strategy employed to generate random sequences was to create four intervals between 0 and 1 with its divisions strategically delimited to create ranges proportional to the probabilities/percentages of the four nucleotides obtained in the positive data set. Then, we

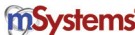

obtain a random number between 0 and 1 and check the interval this value belongs to pick a given random nucleotide. Our code (with a step-by-step guide to using) is available on GitHub: https://github .com/MuriloACassiano/papers-methodology/blob/master/rand_create.ipynb.

**Defining the metrics for promoter analysis.** The true promoter (positive) and random (negative) data sets were used to measure the current tools' capacity to make the correct identification of promoter sequences (it is important to emphasize that we are not using our data sets to retrain and test each methodology). Thus, the results were evaluated comparing the accuracy and Matthews correlation coefficient (MCC) (57), calculated as the following equations:

$$\text{Accuracy} = \frac{(TP + TN)}{(TP + TN + FP + FN)} \tag{1}$$

$$\text{MCC} = \frac{(TP \times TN) - (FO \times FN)}{\sqrt{(TP + FP)(TP + FN)(TN + FP)(TN + FN)}} \tag{2}$$

where $TP$ (true positive) is the number of natural sequences classified as promoters, $TN$ (true negative) is the number of random sequences classified as nonpromoters, $FP$ (false positive) is the number of random sequences classified as promoters, and $FN$ (false negative) is the number of natural sequences classified as nonpromoters. We adopted MCC because it is a metric that deals with unbalanced data sets (i.e., differences in the number of instances in negative and positive data sets), avoiding biases. It achieves high scores only if $TP$ and $TN$ are high, considering both types of correct classification in a single metric, and it has been shown that for this type of binary classification (e.g., promoter/nonpromoter), it is more efficient and less overoptimistic (58).

Sensitivity and specificity scores were also used to give a sense of correct classification of promoters and nonpromoters and are defined as follows:

$$\text{Sensitivity} = \frac{TP}{TP + FN} \tag{3}$$

$$\text{Specificity} = \frac{TN}{TN + FP} \tag{4}$$

Unlike accuracy, sensitivity, and specificity that range from 0 to 1, MCC ranges from –1 (the worst predictor) to 1 (the best predictor) and 0 corresponds to a "random" predictor. By testing the tools with our synthetic random data set, we can measure whether those tools are overfitting their test data sets, and by testing our positive data set (with strong experimental evidence), we are measuring underfitting, once some of our positive sequences probably have already been used to train the tool's algorithms (24). As some of the tools also predict promoters for other sigma factors, to be able to classify all predictions as correct or wrong, we considered random sequences classified as any sigma class promoter as FP and a sigma 70 sequence classified as any other class of sigma promoter as FN. This does not mean that a sigma70 promoter classified as another sigma factor cannot respond to this sigma or even to sigma 70, *in vivo*, as we discuss later.

**Data representation.** For data representation, heatmaps were created by using the R package Heatmap.2 (59), with the default method and using the Jaccard distance method to deal with our binary characteristic vector of 1 (correctly classified) and 0 (wrongly classified) obtained from the tools' results. The Venn diagrams were made by using the Python library matplotlib-venn (https://pypi.org/project/matplotlib-venn/). The logos of count matrices, probability matrices, position weight matrices, and information matrices were constructed by using Logomaker Python library (60). As every result generated by the tools has different formats, these were preprocessed using a text editor or *ad hoc* Python scripts. The data sets used are available for download as files in the supplemental material.

## SUPPLEMENTAL MATERIAL

Supplemental material is available online only.

**FIG S1**, TIF file, 1 MB.

**DATA SET S1,** TXT file, 0.1 MB.

**DATA SET S2,** TXT file, 0.1 MB.

## ACKNOWLEDGMENTS

We thank lab colleagues for insightful comments on this work and María Eugenia Guazzaroni for comments on the final version of the manuscript.

This work was supported by the Sao Paulo Research Foundation (FAPESP, awards 2012/22921-8 and 2019/15675-0). M.H.A.C. was supported by a FAPESP Fellowship (award 2019/06672-7).

We declare that we have no competing interests.

M.H.A.C. and R.S.R. conceived the work. M.H.A.C. performed the analysis. M.H.A.C. and R.S.R. analyzed the results and written the manuscript. All authors read and approved the final manuscript.

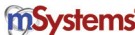

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
