## [Reviewer comments · mSystems]

Benchmarking available bacterial promoter prediction tools: potentialities and limitations

Murilo Henrique Cassiano and Rafael Silva-Rocha

Corresponding Author(s): Rafael Silva-Rocha, FMRP-USP

Review Timeline:

Submission Date:	May 14, 2020
Editorial Decision:	July 12, 2020
Revision Received:	July 31, 2020
Accepted:	August 2, 2020

Editor: Anthony Fodor

Reviewer(s): Disclosure of reviewer identity is with reference to reviewer comments included in decision letter(s). The following individuals involved in review of your submission have agreed to reveal their identity: Zhengchang Su (Reviewer #2)

Transaction Report:

DOI: <https://doi.org/10.1128/mSystems.00439-20>

July 12, 2020

Dr. Rafael Silva-Rocha
FMRP-USP
Cell and Molecular Biology
Av. Bandeirantes 3900
Ribeirão Preto, São Paulo
Brazil

Re: mSystems00439-20 (Benchmarking available bacterial promoter prediction tools: potentialities and limitations)

Dear Dr. Rafael Silva-Rocha:

Your paper has been reviewed by two external reviewers both of which have found substantial merit in the paper. I have also examined the paper and concur with the reviewers that the paper is scientifically strong and is in principle suitable for publication in mSystems. There are, however, a number of minor issues which will need to be addressed before the paper can be accepted for publication. Please see reviewer comments below. As reviewer #2 notes, there are a large number of grammatical errors in the manuscript that need to be corrected. Please make sure that revised version is carefully copy edited before re-submission. Also, please make sure that any custom scripts that are used for the analysis are made available so that others can fully reproduce your work.

To submit your modified manuscript, log onto the eJP submission site at <https://msystems.msubmit.net/cgi-bin/main.plex>. If you cannot remember your password, click the "Can't remember your password?" link and follow the instructions on the screen. Go to Author Tasks and click the appropriate manuscript title to begin the resubmission process. The information that you entered when you first submitted the paper will be displayed. Please update the information as necessary. Provide (1) point-by-point responses to the issues raised by the reviewers as file type "Response to Reviewers," not in your cover letter, and (2) a PDF file that indicates the changes from the original submission (by highlighting or underlining the changes) as file type "Marked Up Manuscript - For Review Only."

Due to the SARS-CoV-2 pandemic, our typical 60 day deadline for revisions will not be applied. I hope that you will be able to submit a revised manuscript soon, but want to reassure you that the journal will be flexible in terms of timing, particularly if experimental revisions are needed. When you are ready to resubmit, please know that our staff and Editors are working remotely and handling submissions without delay. If you do not wish to modify the manuscript and prefer to submit it to another journal, please notify me of your decision immediately so that the manuscript may be formally withdrawn from consideration by mSystems.

To avoid unnecessary delay in publication should your modified manuscript be accepted, it is important that all elements you upload meet the technical requirements for production. I strongly recommend that you check your digital images using the Rapid Inspector tool at <http://rapidinspector.cadmus.com/RapidInspector/zmw/>.

Sincerely,

Anthony Fodor

Editor, mSystems

Journals Department
Reviewer comments:

Reviewer #1 (Comments for the Author):

The authors carry out a systematic comparison of several models (binary classifiers) for bacterial promoters to determine their accuracy, specificity, and sensitivity, using a large, positive dataset of experimentally confirmed E. coli promoters and a large, negative dataset containing randomly generated sequences. The authors provide a very good description of each model, illustrating differences in architecture (neural networks vs. linear discriminant analysis vs. ensemble models vs. support vector machines), features (sequence only vs. physiochemical attributes vs. sequence & physiochemical attributes), and experimental training-testing datasets. The overall analysis is comprehensive, utilizing several metrics and highlighting differences in false positive rates vs. false negative rates. The conclusions are useful; the authors find that more recently developed models are more accurate, using CNNs, AdaBoost ensemble models, and SVMs, as compared to the predominantly used classic model (BPROM), which uses linear discriminant analysis. The authors are neutral in their analysis and emphasize that the more accurate models were more recently developed and therefore had access to larger datasets and more sophisticated ML techniques.

As one major deficiency, all of the analyzed models are binary classifiers, and yet we know that transcription can take place promiscuously from randomly generated promoter sequences. It is therefore important to highlight and emphasize that, from the binary classifier's point of view, the definition of a promoter is a sequence that is transcribed at a sufficiently high rate, beyond some critical (but arbitrarily defined) threshold. The authors mention this briefly, but it is a very important point and deserves to be explicitly mentioned in the results with its own paragraph in the discussion

/ conclusions. In order to push forward the state-of-the-art in promoter prediction, it will be important to develop models that can predict a promoter's transcription rate in terms of a continuous-valued number (a regressor, not a classifier). Only then could a model distinguish between promoters with low vs. high transcription rates, even though both produce enough mRNA to affect cellular phenotypes.

Typos:

Line 200. "remining" to "remaining"

Reviewer #2 (Comments for the Author):

The paper describes a comparative study on existing tools for predicting bacterial core promoters around transcription start sites. The experiments were well-designed and the paper was well-structured. The results can be useful for users to choose appropriate tool and for developers to develop more accurate tools.

Some concerns:

1. Page 3, paragraphs 2 and 3 repeat the review of applications of ML in promoter prediction.
2. Negative promoter sequences were constructed with an ad hoc script written in Python, but the underlying algorithm should also be described that guarantees the positive and negative sets have same nucleotides frequency distribution.
3. The paper contains many grammatical errors: the following are just some of them:

Line 46: "sensibility" appears to be "sensitivity";

line 60: "required for its transcription " should be "required for their transcription";

line 81: "position-weighted matrices" appears to be "position-weight matrices";

Line 84: "such as sequence alignment-base kernel" should be "such as sequence alignment-based kernel ";

Line 89: "other methods that includes" should be "other methods that include ";

Line 99: "making difficult to compared novel emerging alternatives" should be "making difficult to compare novel emerging alternatives ";

Line 99: "the current state of art" should be "the current state of the art";

Line 129: "distance and used to training a neural network" appears to be "distance and used to train a neural network";

Line 135: "random selected intergenic regions" appears to be "randomly selected intergenic regions";

Line 203: "followed by iPromoter-2L (83.8% TP), 203 70ProPred (89.7% TP), iPro70-FMWin (94.5% TP) and MulTiPly (81.2% TP)": the order is not in the supposed decreasing order;

Line 217: "573 sequences (62.2%) where correctly classified by all 5 algorithms " appears to be "573 sequences (62.2%) were correctly classified by all 5 algorithms";

Line 231: "we analyze the information content " appears to be "we analyzed the information content";

Line 232: "top 5 tools analyses here" appears to be "top 5 tools analysed here";

Line 232: "Logo on Figure 4 and 5 for positive and negative dataset

" appears to be "Logos on Figures 4 and 5 for the positive and negative datasets"

Line 251: "highlighting the importance of choosing appropriated non-promoter sequences

" appears to be "highlighting the importance of choosing appropriate non-promoter sequences";

Line 258: "attributes derivate from " appears to be "attributes derived from" ;

"

Line 271 "could be incorporate in prediction methods" appears to be "could be incorporated in prediction methods ";

Line 272 "interrelation between the UP element and a subunit of RNAP were found to play a role on transcription initiation and promoter activity (44) and switch preference" should be "interrelation between the UP element and a subunit of RNAP was found to play a role in transcription initiation, promoter activity (44) and switch preference".

Dear Editor,

Thanks for the comments on our manuscript. We have now addressed all comments from the reviewers and we are presenting an improved version of the manuscript now. Specially, we made available the code used to generate the negative dataset and corrected the manuscript.

We hope you find this new version suitable for publication at mSystems. Sincerely yours,

Rafael

Reviewer 1

The authors carry out a systematic comparison of several models (binary classifiers) for bacterial promoters to determine their accuracy, specificity, and sensitivity, using a large, positive dataset of experimentally confirmed *E. coli* promoters and a large, negative dataset containing randomly generated sequences. The authors provide a very good description of each model, illustrating differences in architecture (neural networks vs. linear discriminant analysis vs. ensemble models vs. support vector machines), features (sequence only vs. physiochemical attributes vs. sequence & physiochemical attributes), and experimental training-testing datasets. The overall analysis is comprehensive, utilizing several metrics and highlighting differences in false-positive rates vs. false-negative rates. The conclusions are useful; the authors find that more recently developed models are more accurate, using CNNs, AdaBoost ensemble models, and SVMs, as compared to the predominantly used classic model (BPRM), which uses linear discriminant analysis. The authors are neutral in their analysis and emphasize that the more accurate models were more recently developed and therefore had access to larger datasets and more sophisticated ML techniques.

As one major deficiency, all of the analyzed models are binary classifiers, and yet we know that transcription can take place promiscuously from randomly generated promoter sequences. It is therefore important to highlight and emphasize that, from the binary classifier's point of view, the definition of a promoter is a sequence that is transcribed at a sufficiently high rate, beyond some critical (but arbitrarily defined) threshold. The authors mention this briefly, but it is a very important point and deserves to be explicitly mentioned in the results with its own paragraph in the discussion/conclusions. In order to push forward the state-of-the-art in promoter prediction, it will be important to develop models that can predict a promoter's transcription rate in terms of a continuous-valued number (a regressor, not a classifier). Only then could a model distinguish between promoters with low vs. high transcription rates, even though both produce enough mRNA to affect cellular phenotypes.

R: Thanks for this comment. This is true, all models available are only binary classifiers unable to predict promoter strength. The main reason for this is that in fact promoters are almost never in isolation (as in synthetic circuits) but are composed by the core element (-35/-10) and some *cis*-regulatory elements. If not all *cis*-elements are known (which is not uncommon), a promoter can be miss interpreted as constitutive and this may create a biases during promoter identification. Therefore, it is not simple yet to assign true promoter strength to a core promoter sequence unless all interacting elements are known. We added to the text a paragraph discussing some approaches that have been suggested to predict promoter strength and their limitations.

Typos:

Line 200. "remining" to "remaining"

R: We corrected the manuscript for this and other errors.

Reviewer 2

The paper describes a comparative study on existing tools for predicting bacterial core promoters around transcription start sites. The experiments were well-designed and the paper was well-structured. The results can be useful for users to choose appropriate tool and for developers to develop more accurate tools.

Some concerns:

1. Page 3, paragraphs 2 and 3 repeat the review of applications of ML in promoter prediction.

R: Thanks for the comments. In paragraph 3 we describe in more detail, as reliable as we could, the approaches of each tool we found, while in paragraph 2 we mention the ML application in an introductory manner.

2. Negative promoter sequences were constructed with an ad hoc script written in Python, but the underlying algorithm should also be described that guarantees the positive and negative sets have same nucleotides frequency distribution.

R: We added the explanation of the algorithm in the Methods section, and now we added a link to our code on GitHub.

3. The paper contains many grammatical errors: the following are just some of them:

Line 46: "sensibility" appears to be "sensitivity";

R: Corrected.

line 60: "required for its transcription " should be "required for their transcription";

R: Corrected.

line 81: "position-weighted matrices" appears to be "position-weight matrices";

R: Corrected.

Line 84: "such as sequence alignment-base kernel" should be "such as sequence alignment-based kernel";

R: Corrected.

Line 89: "other methods that includes" should be "other methods that include";

R: Corrected.

Line 99: "making difficult to compared novel emerging alternatives" should be "making difficult to compare novel emerging alternatives";

R: Corrected.

Line 99: "the current state of art" should be "the current state of the art";

R: Corrected.

Line 129: "distance and used to training a neural network" appears to be "distance and used to train a neural network";

R: Corrected.

Line 135: "random selected intergenic regions" appears to be "randomly selected intergenic regions";

R: Corrected.

Line 203: "followed by iPromoter-2L (83.8% TP), 203 70ProPred (89.7% TP), iPro70-FMWin (94.5% TP) and MulTiPly (81.2% TP)": the order is not in the supposed decreasing order;

R: Corrected.

Line 217: "573 sequences (62.2%) where correctly classified by all 5 algorithms " appears to be "573 sequences (62.2%) were correctly classified by all 5 algorithms";

R: Corrected.

Line 231: "we analyze the information content " appears to be "we analyzed the information content";

R: Corrected.

Line 232: "top 5 tools analyses here" appears to be "top 5 tools analysed here";

R: Fixed.

Line 232: "Logo on Figure 4 and 5 for positive and negative dataset " appears to be "Logos on Figures 4 and 5 for the positive and negative datasets"

R: Corrected.

Line 251: "highlighting the importance of choosing appropriated non-promoter sequences " appears to be "highlighting the importance of choosing appropriate non-promoter sequences";

Line 258: "attributes derivate from " appears to be "attributes derived from" ;

R: Corrected.

Line 271 "could be incorporate in prediction methods" appears to be "could be incorporated in prediction methods";

R: Corrected.

Line 272 "interrelation between the UP element and a subunit of RNAP were found to play a role on transcription initiation and promoter activity (44) and switch preference" should be "interrelation between the UP element and a subunit of RNAP was found to play a role in transcription initiation, promoter activity (44) and switch preference".

R: Corrected.

August 2, 2020

Dr. Rafael Silva-Rocha
FMRP-USP
Cell and Molecular Biology
Av. Bandeirantes 3900
Ribeirão Preto, São Paulo
Brazil

Re: mSystems00439-20R1 (Benchmarking available bacterial promoter prediction tools: potentialities and limitations)

Dear Dr. Rafael Silva-Rocha:

Your manuscript has been accepted, and I am forwarding it to the ASM Journals Department for publication. For your reference, ASM Journals' address is given below. Before it can be scheduled for publication, your manuscript will be checked by the mSystems senior production editor, Ellie Ghatineh, to make sure that all elements meet the technical requirements for publication. She will contact you if anything needs to be revised before copyediting and production can begin. Otherwise, you will be notified when your proofs are ready to be viewed.

In reading over the manuscript, I noted a fair number of minor grammatical errors. These have been pasted at the end of this e-mail. Please make sure that these are fixed before final publication.

Sincerely,

Anthony Fodor
Editor, mSystems

Journals Department
Supplemental Material 1: Accept
Figure S1: Accept
Supplemental Material 2: Accept

37 : massive -> massively parallel

39: have become -> having become

39: there is no systematic comparison of such tools -> no systematic comparison of such tools has been performed

49: Therefore, we exploit here -> We present here

50: hope future works can be built upon -> we hope future work can build upon

51: such quite useful -> this useful

111: Below, we described -> Below, we describe

120: past the sequences -> paste the sequences

122: and its position -> and their positions

138: then past the sequences -> then paste the sequences

141: authors took the -> the authors took the

182: MULTiPly (41) web application -> The MULTiPly (41) web application

189: which class of sigma promoter it belongs -> to which class of sigma promoter it belongs

191-192: have a similar way to use -> are used in similar ways

239: fuzzy signal -> fuzzier signal

247: could reveal that these tools are -> might be explained by these tools

264: well-validate -> well-validated

271: several prior information -> several sources of prior information

281: a large number of elements, making the transcription -> a large number of elements and therefore the transcription

291: also added considered variance -> also added variance

304: promoter sequence, by using cross-validation -> promoter sequence, and by using cross-validation

309: this energy matrices -> these energy matrices

310: composed by one -> composed of one

320: community -> the community

332: we selected it to posterior -> we selected it for posterior

342: Remove the (?)

350: there are no -> there is no

388: those tools have overfitting with their test datasets -> those tools are overfitting their test datasets

618: lack in experimental -> lack of experimental